

**Landslides distribution at tributaries with different evolution stages in Jiangjia Gully,**
2                                      **southwestern China**

3             Xia Fei Tian [a, b, c], Yong Li [a, b*], Quan Yan Tian [c, d], Feng Huan Su [a, b]

a.   Key Laboratory of Mountain Hazards and Surface Process, CAS, Chengdu, 610041, China
b.   Institute of Mountain Hazards and Environment, CAS, Chengdu, 610041, China
c.   University of Chinese Academy of Sciences, Beijing 100049, China
d.   Cold and Arid Region of Environmental and Engineering Research Institute, CAS
*Corresponding author: Li Yong
E-mail: ylie@imde.ac.cn
**Abstract:** Jiangjia Gully (JJG) is known for its high frequency and variety of debris flows, especially
the intermittent surges of various flow regimes and materials. Observation indicates that the surges
come from various tributaries with different landslides activities. In this study, 81 tributaries of JJG are
taken from DEM with 10 m grid cells, and the hypsometric curves are used to characterize their
evolution stages; five stages are identified by the evolution index (EI, the integral of the hypsometric
curves) and most tributaries are in relative youth stage with EI between 0.5 and 0.6. Then 908
landslides are interpreted from Quickbird satellite image of 0.61 m resolution, and it is found that LD
(LD = landslides number in a tributary/ the tributary area) increases exponentially with EI, while $LA_p$
( $LA_p$ = landslides area in a tributary/ the tributary area) fluctuates with EI, meaning that landslides are
inclined to occur in tributaries with EI between 0.5 and 0.6, and thus these tributaries are the main
material sources supplying for debris flows.
**Key words:** Hypsometric curve; Evolution stages; Tributaries; Landslides distribution

**1 Introduction**

Geomorphic evolution has been one of the important research topics in geomorphology,

hypsometric analysis has been used to deal with erosional topography and the process of landform
development (Bartolini et al., 2003; Li et al., 2011; Lv et al., 2005). Strahler (1952) asserted that
different types of landform have different characteristic shape of their hypsometric curves, dividing



landform into 'young' and 'mature' with the hypsometric integral decreasing. In this meaning, the
integral can be defined as the evolution index (EI) of the tributary. Meanwhile, the hypsometric curves
are related to tributary form and erosional process, and are used to interpret landform development
stages (Schumm, 1956; Strahler, 1952, 1957). In addition, the relationship between EI and tributary
characteristics changes with scales. For example, the dissection index of tributaries presents various
relationships to EI depending on scale of the tributaries. For the 5th-order tributaries, their correlation
is r = 0.41, whereas for the 4th-order, it is r = 0.24, and it becomes negative correlation for the
3rd-order (Hamza V et al., 2018). Combined with the results of field investigation, this study adopts the
tributary scale that debris flow easily occurs to meet our research needs.

For a given watershed, especially a small gully in mountains (below 100 km$^2$ and most below 10

km$^2$), the tributaries with different EI present various topographic characteristics. Similarly, significant
difference exists in the distribution of landslides among various tributaries, landslides are frequent in
some tributaries while occasional in others (Baum et al., 2005; Pradhan and Sameen, 2017; Wang et al.,
2006; Wieczorek, 1996). Although landslides distribution per se are influenced by many factors, such
as lithology, topographic characteristics (Langebein and Basil, 2007) and climatic variable (Huggel et
al., 2005; Wieczorek and Glade, 2005), and the specific evolution state of a tributary is the result
affected by these comprehensive factors as well. Therefore, the relationship between EI and landslides
distribution has special significance to reveal the landslides distribution characteristics with regards to
mountainous tributaries, which, however, has gotten little attentions in literatures.

In this paper, a case study is conducted in Jiangjia Gully (JJG), where weak and similar lithology,

disparate topography, sparse vegetation, and unconsolidated deposits are widely distributed in
tributaries. In addition, it is known for the high variety of debris flows; each debris-flow event consists
of tens or hundreds surges of different flow regimes, velocities, discharges, and total volumes (Li et al.,
2015; Li et al., 2013; Arai, 2017). In particular, the surges are composed of different materials,
suggesting that they come from different sources (Xiang et al., 2015). In other words, each debris flow
in JJG comes from different tributaries (Bollschweiler et al., 2007; Li et al., 2012; Li et al., 2013; Li et
al., 2015). Moreover, the debris flow behaviors in JJG are representative and similar phenomena are
subsistent in other parts of the world. Generally, the flow surges are originated from different tributaries
and the material supplies are mainly from landslides (including avalanches, soil failures and other slope
processes) (Beguería, 2006). So the study of landslides distribution in different evolution stages is of



great significance to reveal the landslides distribution characteristics of the tributaries with similar
lithology and disparate topography, but also can roughly determine the material supply and explain the
formation mechanism of debris flow surges.
**2 Study area and data collection**
2.1 Setting of the study area
JJG is located in the Xiaojiang River of the Upper Changjiang River. The mainstream channel
length is $1.39 \times 10^4$ m and the gully area is $4.84 \times 10^7$ m$^2$ (Fig. 1). This region undergoes active
neotectonic movement, faults, and folds; and rocks are dominated by slate, dolomite, limestone, basalt
and breccia rocks, which are easily weathered (Gabet and Mudd, 2006). Generally, weak lithology,
wide faults and sparse vegetation are the obvious characteristics of the gully, and the tributaries are in
steep topography and intense landslide activity, with wide distribution of quaternary unconsolidated
deposits. Loose materials are widely distributed in the gully and debris flows occur frequently, which
are the major material sources for the debris flows. According to the statistics data, the landslides area
reaches 16.4 km$^2$ that accounts for 39.7% of the gully area. As well, average annual sediment yield by
debris flow is about $1.54 \times 10^6$ m$^3$ (Wu et al., 1990; Zhuang et al., 2015).



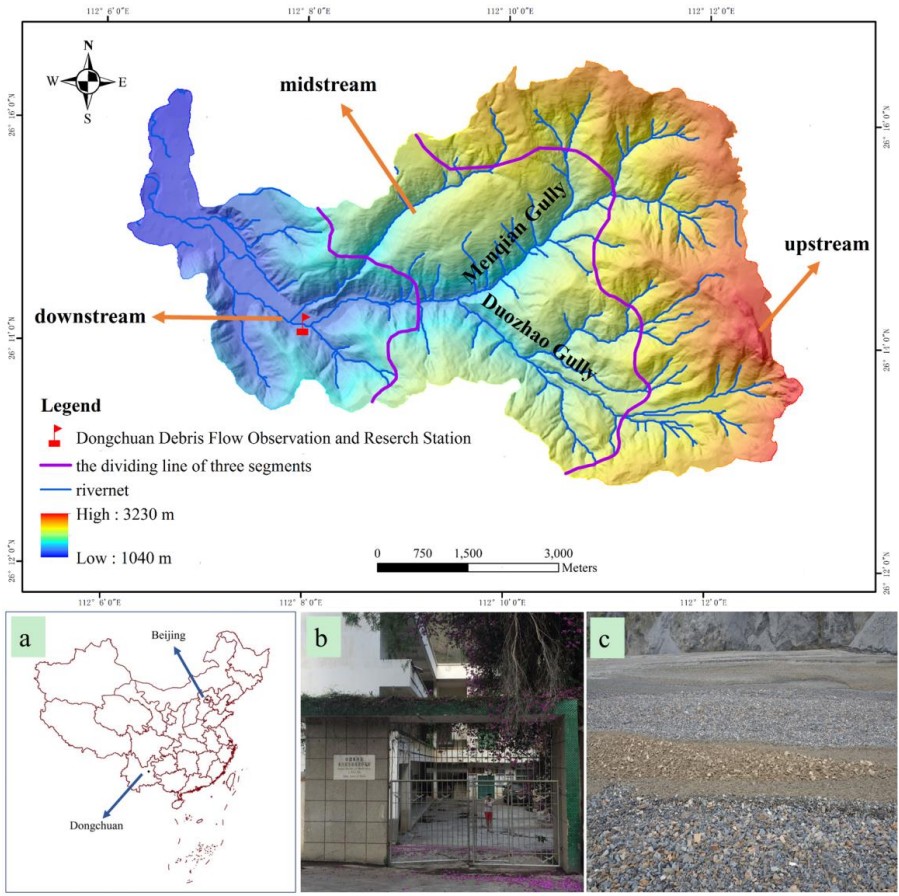


**Fig. 1** The location of JJG. a The location of Dongchuan in China. b Dongchuan Debris Flow

Observation and research stations. c Deposition of surges.

The most remarkable features of debris flow in JJG are the high frequency in occurrence and great

variety of flow regime and magnitude. Each occurrence contains tens to hundreds of surges (Li et al.,
2012); the surges are separated in time and space, and different from one another in density, velocity,
and sediment concentration. The variation of flow velocity with density is shown in Fig. 2, which
contains surges in one single event on July 12, 2017.

The great variety of surges densities, with different material compositions, can be attributed to

different sources; this means that even a single surge material comes from different tributaries in most
cases (Webb et al., 1989). As observed in the last decades, debris flows almost come from the north
branch, the Menqian Gully, while the south branch, the Duozhao Gully, is often silent. This presents the
gross distinction of material and landslides activities in JJG, which further implies that there must be





more differences in tributaries.

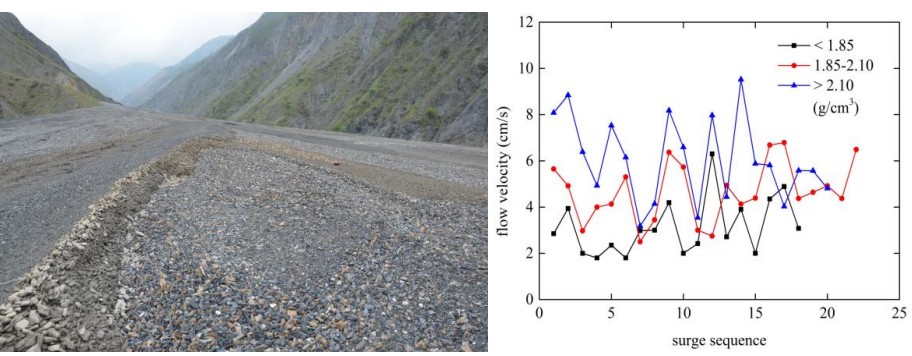


**Fig. 2** Debris flow surges deposit in the mainstream of JJG.

2.2 Data collection
2.2.1 The tributaries divided in JJG
Digital elevation model with spatial resolution 10 m is used in this study to generate elevation and
area information, and 81 tributaries are divided into. The tributaries are divided based on field
investigation that debris flow easily occurs. Some tributaries in field are displayed in Fig. 3, obviously,
there are significant differences in tributaries characteristics, and the study of such tributaries size is of
great significance to the formation and occurrence of debris flows. The tributary area varies between
$8.7 \times 10^4 \sim 2.07 \times 10^6$ m$^2$ and cover a total area of $4.62 \times 10^7$ m$^2$, about 95% of the whole gully. The
serial number of tributary in subregions is presented in Table 1.





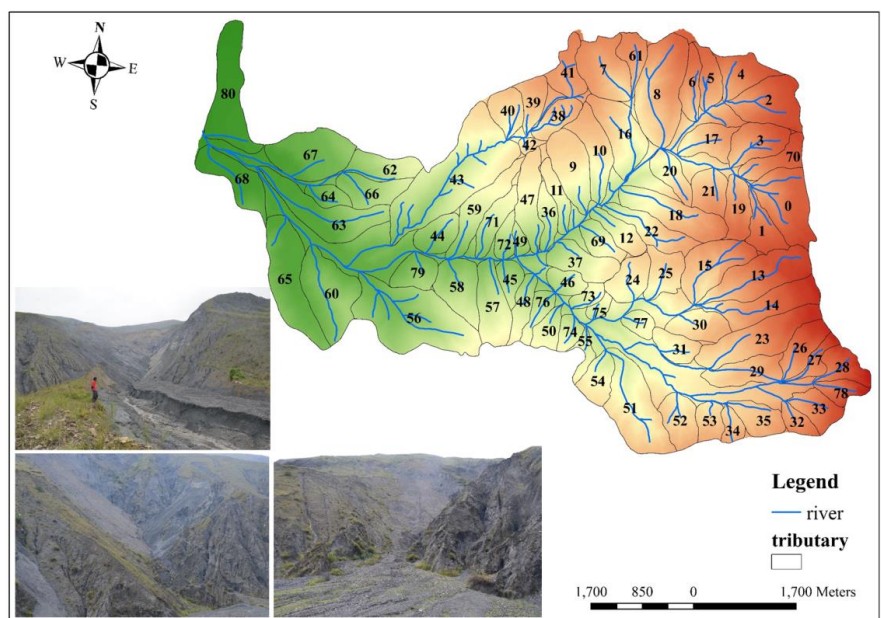


**Fig. 3** The tributaries divided of JJG. Some tributaries in the field are shown on the map.
**Table 1** The tributaries distribution in subregions.

| Subregion | The No. of tributaries |
|-----------|------------------------|
| Menqian Gully | 2, 4, 5, 6, 7, 8, 9, 10, 11, 12, 16, 17, 18, 20, 22, 36, 37, 47, 49, 61, 69 |
| Duozhao Gully | 13, 14, 15, 23, 24, 25, 26, 27, 28, 31, 31, 32, 33, 34, 35, 45, 46, 48, 50, 51, 52, 53, 54, 55, 73, 74, 75, 76, 77, 78 |
| Upstream | 0, 1, 2, 3, 4, 5, 6, 7, 8, 9, 10, 13, 14, 15, 16, 17, 19, 20, 21, 23, 26, 28, 29, 30, 32, 33, 34, 35, 38, 39, 40, 41, 42, 61, 70 |
| Midstream | 9, 11, 12, 22, 31, 36, 37, 43, 45, 46, 47, 48, 49, 50, 51, 54, 55, 59, 69, 71, 72, 73, 74, 75, 76, 77 |
| Downstream | 44, 56, 58, 60, 62, 63, 64, 65, 66, 67, 68, 79, 80 |

2.2.2 The hypsometric curve and EI
Hypsometric curve for each tributary is calculated. The hypsometric curve is generated by plotting
the relative area along the abscissa and the relative height along the ordinate. The relative height can be
obtained as the ratio of the height of a given contour (h) from the base plane of the stream mouth to
total height of the tributary with reference to the maximum elevation (H), and the relative area is
obtained as the ratio of the area above a particular contour (a) to the total area of the tributary



encompassing the outlet (A) (Strahler, 1952).

Hypsometric integral is the area between the hypsometric curve (y=h/H and x=a/A) and

coordinate axis (Strahler, 1952, 1957), which can be defined as the evolution index (EI).
2.2.3 The extraction of landslides information

Quickbird image of 0.61 m resolution is purchased to create an inventory of landslides. The

satellite image is adopted in this study with low cloud shadow coverage, and the aerial coverage of the
cloudy area is 0.09 km$^2$ in the study area, about 0.18% of the gully. The atmospheric correction and
radiometric correction have been carried out by using the calibration function within the tools of Envi
5.1 software, and 4, 3, 2 bands are combined to false color image stretched of contrast using standard
deviation method. Both landslides number and landslides area are necessary to interpret, so the equal
area projection is adopted, which has less impact on the landslides area. The landslides information
becomes easily extracted on the source image after processing, which is beneficial to the work of visual
interpretation, and thus ensures the accuracy of the results.

Landslides are mapped from high resolution satellite data acquired using visual image

interpretation on Arc GIS 10.3 software with false color composites or panchromatic images uniformly
on 1:5000 scale. The individual landslides initiation zones are indicated using polygons. In the case of
complex situations where many landslides are interconnected, it is difficult to identify the individual
initiation zones. Use of high resolution images enables demarcation of clustered landslides as
individual polygons. The minimum size of landslides area is determined as 0.38 m$^2$ and the area below
this value are not considered as the resolution of the satellite image is not sufficient to extract
landslides information preferably.

In the interpretation process, we make use of the following diagnostic features: the tone, texture,

pattern and shape or form. Meanwhile, direct method, comparison method, integrated reasoning
method and other synthetical methods are always used (Dai and Lee, 2002; Kumar et al., 2017;
Valenzuela et al., 2017). Using the methods above, 908 landslides have been identified, with area
ranging of $3.8 \times 10^{-1} \sim 6.7 \times 10^5$ m$^2$. In addition, fieldwork was carried out in May and June 2017. We
investigated distribution of 100 landslides with the GPS instrument, and the accuracy achieves 89.21%.
The data is used to analyze the relationship between EI with LA$_p$ and LD of each tributary, of which
LA$_p$ is landslides area in a tributary/ the tributary area (%) and LD is landslides number in a tributary/
the tributary area (/$10^6$ m$^2$).





**3 Evolution division of JJG**
3.1 Hypsometric analysis
The hypsometric curves for tributaries are shown in Fig. 4:

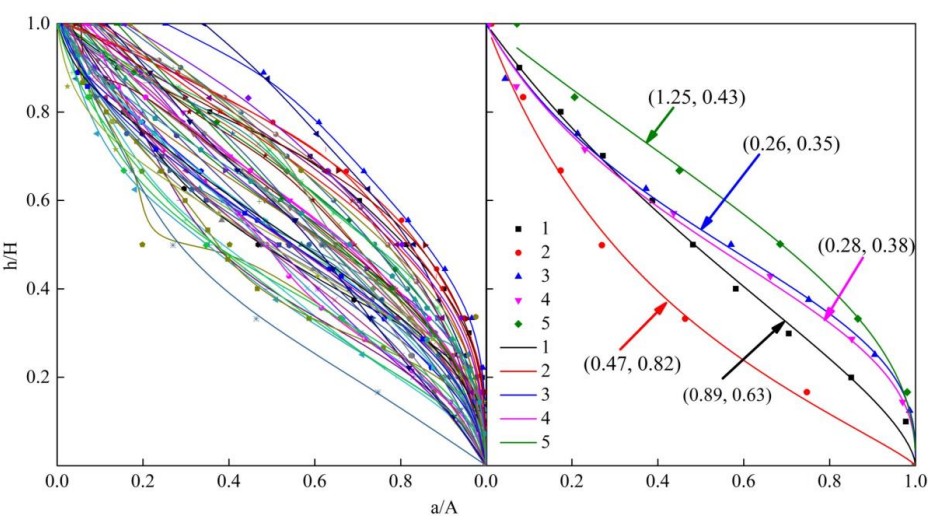


**Fig. 4** The hypsometric curves of different tributaries.
The curves present various types, such as convex, concave and others between them. The curves
can be fitted by:
$$y^{1/n} = k(1-x)/(x+k) \qquad (2)$$
Where k and n are parameters. The fitting coefficient $R^2$ achieves 90% above. It is found that higher the
curve is, greater the k is. Meanwhile, Fig. 4 shows that with the rising of curves, n is decreasing.
3.2 Evolution division of JJG
Then the EI of each tributary in JJG is calculated, which varies from 0.32 to 0.84. According to
Strahler, there are three stages: inequilibrium or youthful stage (EI > 0.6), equilibrium or mature stage
(EI between 0.3 and 0.6), and monadnock or old stage (EI < 0.3) (Strahler 1952). In order to distinguish
the evolution differences of the tributaries, we conduct a more detailed classification and the EI of
tributaries in JJG are divided into five groups (Fig. 5):
I (< 0.45), appears in downstream areas and near the outlet of the gully;
II (0.45-0.55), occurs mostly in the Duozhao Gully;
III (0.55-0.65), mostly distributes in midstream and downstream;
IV (0.65-0.75), mostly in midstream and upstream;
V (>0.75), mainly distributes in the headwaters of the Menqian Gully;

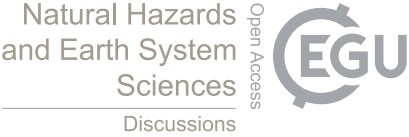

Moreover, it is found that the EI satisfies the Weibull distribution with the scale parameter of 0.58

and the shape parameter of 6.08 (Fig. 6), the small value of scale parameter means that EI is much
concentrated and EI of most tributaries in JJG is between 0.5 and 0.6. According to the frequency
distribution of EI, the tributaries of JJG is generally in mature and youthful evolution stages, that is the
reason why high frequency debris flow occurred in JJG in the past several decades.

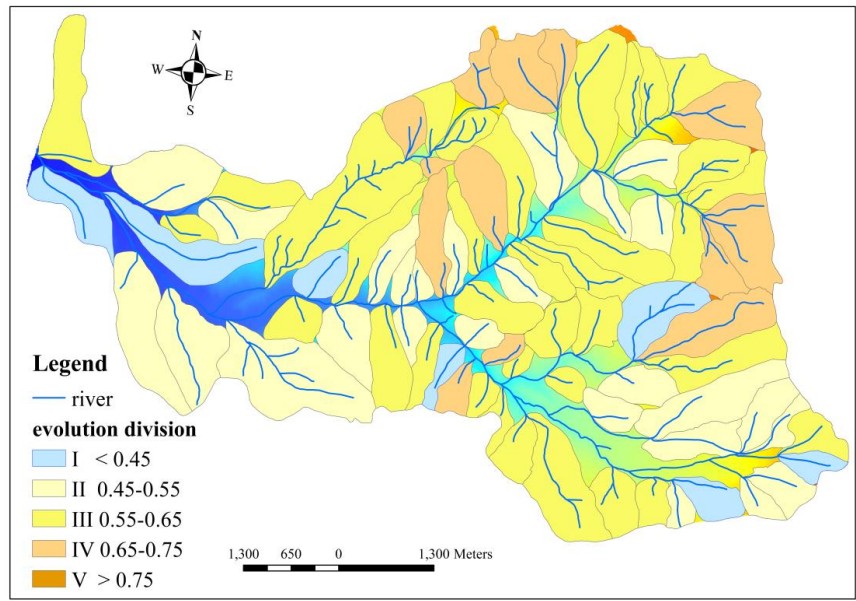


**Fig. 5** The evolution division and EI distribution of tributaries in JJG.

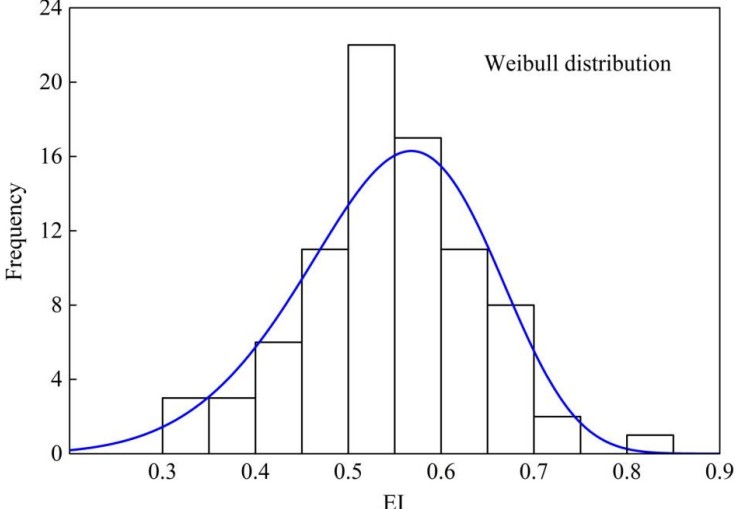






**Fig. 6** The EI frequency distribution of tributaries in JJG.
3.3 Hypsometric curves of different evolution stages
For more clear, we display the hypsometric curves of different evolution stages in Fig.7; in
particular, the inflection points of the curves (the rectangle in each plot) are displayed in different
position of the curves, they can reflect the geomorphologic features of the tributaries. The inflection
point indicates the elevation of a tributary with area varing. When the point is high, the changing
occurs at the high elevation, i.e., mainly in the upstream of the tributary.
Obviously, curves in different evolution stages exhibit different characteristics. The bigger the
evolution index is, the higher the inflection points of the curves are. In addition, the distribution of
inflection points displays distinct spatial features in different evolution stages (Fig. 7).

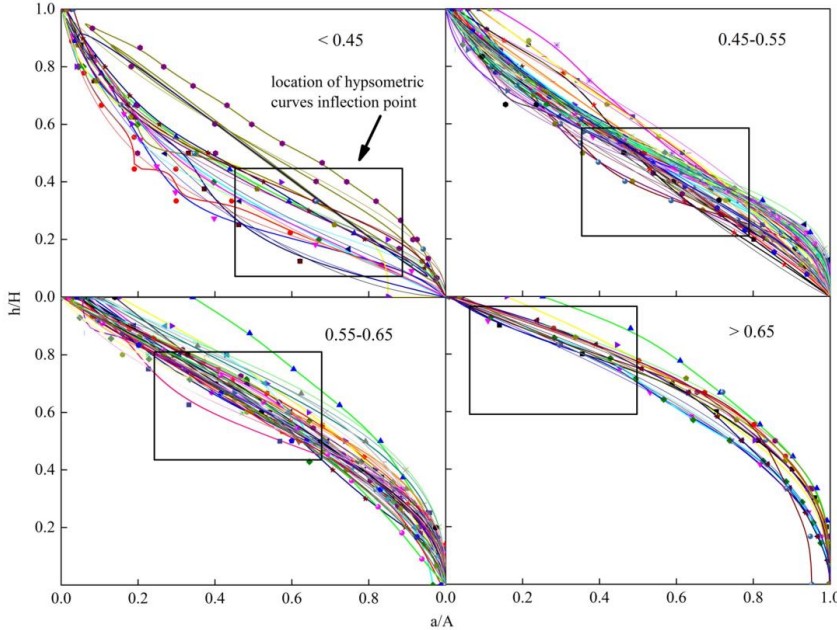


**Fig. 7** Hypsometric curves and the position of inflection points in different evolution stages.
Corresponding to the point h is the area a(h), which is the area in tributary above the inflection
point. For a given elevation of point, larger area above it means strong slope process in the upstream.
For example, inflection points in EI between 0.45~0.55 are generally higher than those in EI below
0.45, indicating that such tributaries are more prone to landslides activities. Correspondingly, the lower
the hypsometric curve is, the more concave the curve is presented, and the smaller of the elevation
value corresponding to the inflexion point is, which indicates that the elevation changing in unit area is


small, such a tributary is not conducive to the occurrence of landslides.
**4 Landslides distribution in relation to EI**
4.1 Landslides distribution of JJG
A total of 908 landslides have been identified, with area ranging from $3.8 \times 10^{-1} \, \text{m}^2$ to $6.7 \times 10^{5} \, \text{m}^2$.
The spatial distribution of landslides is shown in Fig. 8.

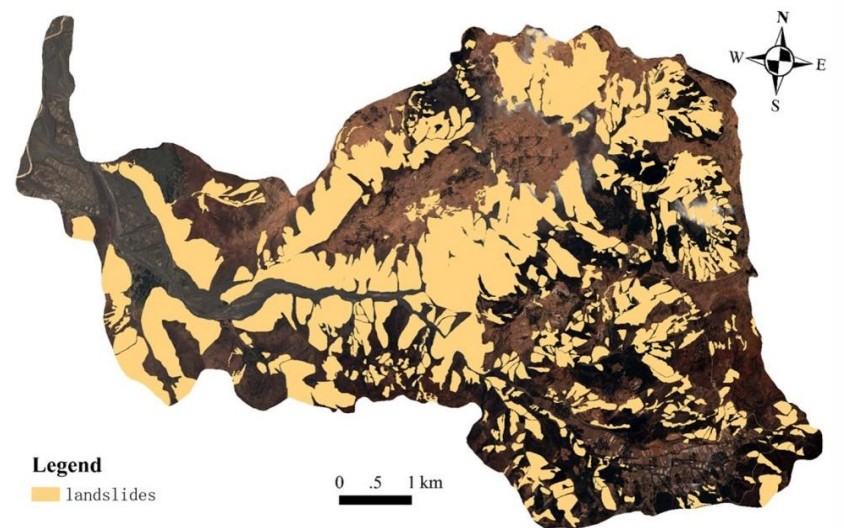


**Fig. 8** Spatial distribution of landslides in JJG.
Landslides are mainly distributed in both sides along the mainstream channels. In details,
landslides in Menqian Gully are more concentrated while in Duozhao they are scattering, which is
consistent with field observations that landslides are always more frequent in clusters in vulnerable
areas.
The landslides distribution in subregions is shown in Table 2, which indicates that $LA_p$ of
Menqian Gully is greater than Duozhao Gully, while LD presents a reverse tendency, which means that
landslides are concentrated and larger-scale in Menqian Gully, while small landslides are scattering in
Duozhao, as seen in Fig. 8. In addition, the $LA_p$ of midstream is greater than upstream and downstream,
and LD in upstream is the biggest.
**Table 2** The landslides distribution in subregions.

| Subregion | The area (km²) | The area percentage (%) | Landslides | | | |
|---|---|---|---|---|---|---|
| | | | LA (km²) | $LA_p$ (%) | LN | LD (km⁻²) |



| | | | | | |
|---|---|---|---|---|---|
| Menqian Gully | 12.52 | 27.12 | 5.67 | 45.32 | 248 | 19.81 |
| Duozhao Gully | 15.01 | 32.52 | 4.62 | 30.81 | 403 | 26.85 |
| Upstream | 19.60 | 42.47 | 6.22 | 31.71 | 526 | 26.83 |
| Midstream | 16.46 | 35.66 | 7.55 | 45.88 | 356 | 21.63 |
| Downstream | 10.71 | 23.21 | 3.50 | 32.71 | 67 | 6.25 |

4.2 Landslides distribution in different evolution division
4.2.1 The landslides distribution related to evolution stages of all tributaries
The evolution division and landslides distribution layers are overlaid to form the spatial
distribution map, as shown in Fig. 9. It is clear that major of landslides are distributed in subregions of
III and IV, with EI between 0.55 ~ 0.75.

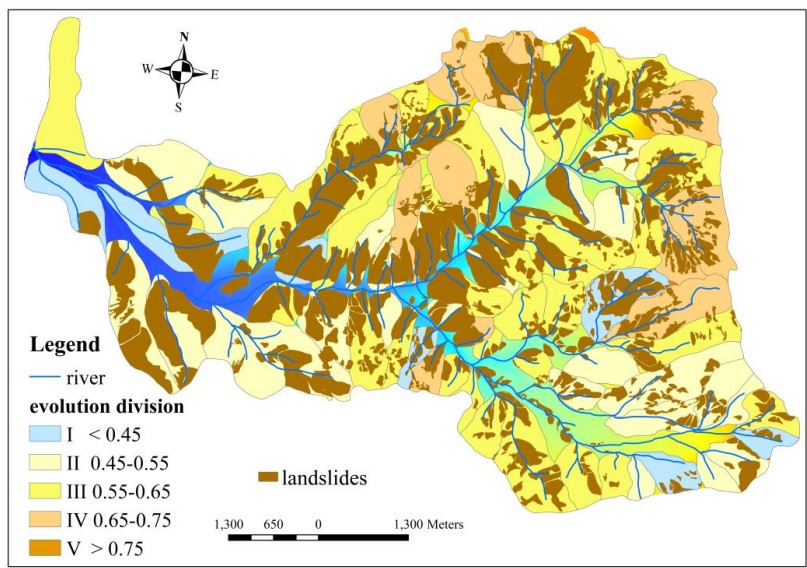


**Fig. 9** Landslides distribution in various evolution stages.

Fig. 10 shows how LD and $LA_p$ vary with EI. It shows that LD increases exponentially with EI
increasing, which means that more landslides occur in the tributaries at younger evolutionary stage.
Meanwhile, the greater fluctuation of $LA_p$ is in tributaries with the range of EI less than 0.54, while a
smaller fluctuation is in tributaries of EI more than 0.54, and the $LA_p$ is generally smaller than other
evolution stages in active evolution stage.





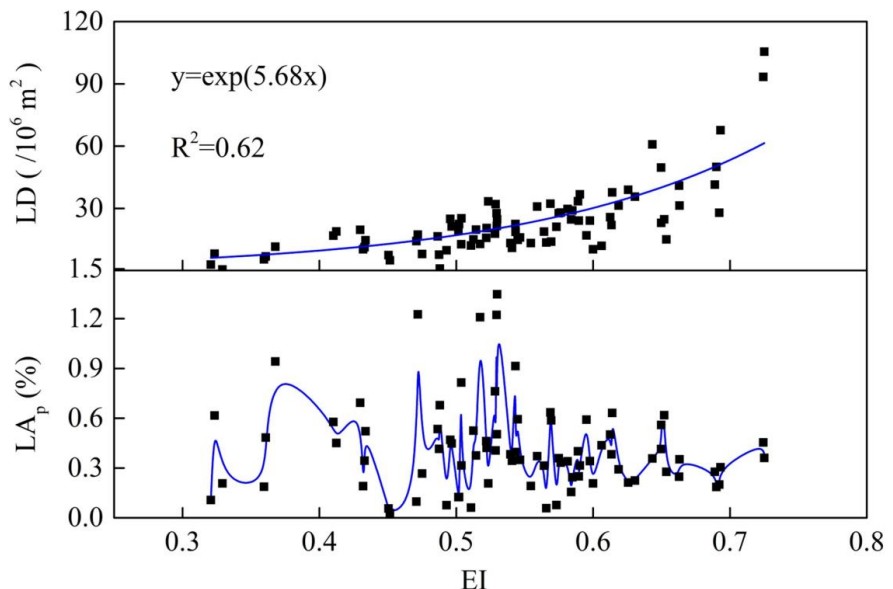

**Fig. 10** Relationship between landslides and EI.

The percentage of $LA_p$ increases at the range of less than 0.52 and decreases at the range of more than 0.52 approximatively. The $LA_p$ is generally greater in the range of EI smaller than 0.55, and $LA_p$ decreases clearly with EI exceeding 0.52. Therefore, more landslides mainly occur in active stages on small scale.

4.2.2 Landslides distribution in typical subregions

The major branches of JJG, the Gully of Menqian and Duozhao, are distinctive in debris flow and landslides activities. As mentioned above, landslides are more scattering in Duozhao and more concentrated in Menqian. Now we consider how landslides distribute in tributaries in these subregions.

Fig. 11 shows that in both gullies LD increases exponentially with EI, almost in the same exponential function. As for $LA_p$, several peaks occur in different EI values in Menqian Gully but only a single peak occurs (around EI with 0.55) in Duozhao Gully, meaning that landslides are widely distributed in tributaries with EI >0.45 in Menqian Gully.




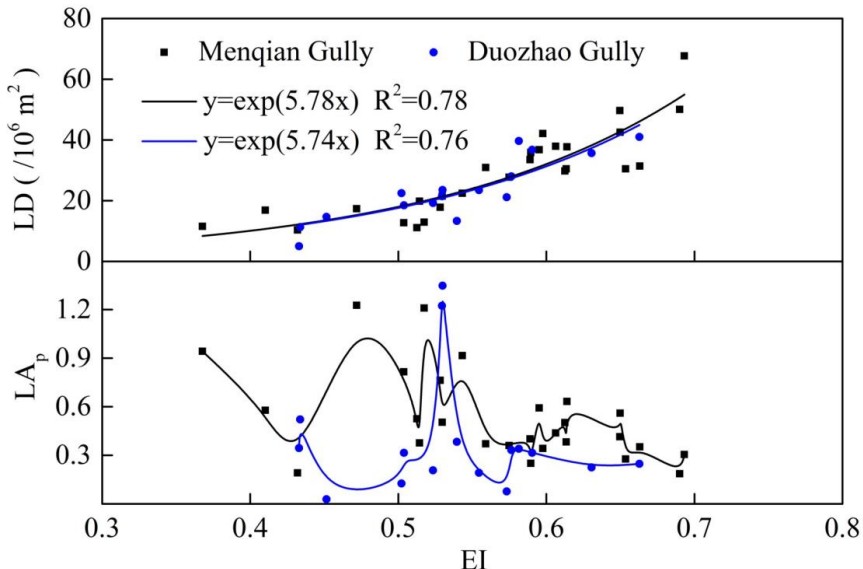


**Fig. 11** Relationship between landslides and EI in typical gullies.

Similarly, we consider LD and $LA_p$ in the regions of the upstream, midstream and downstream in
JJG that have visible terrain difference, as shown in Fig. 12. Again it is found that LD increases
exponentially with EI both in the upstream and midstream.

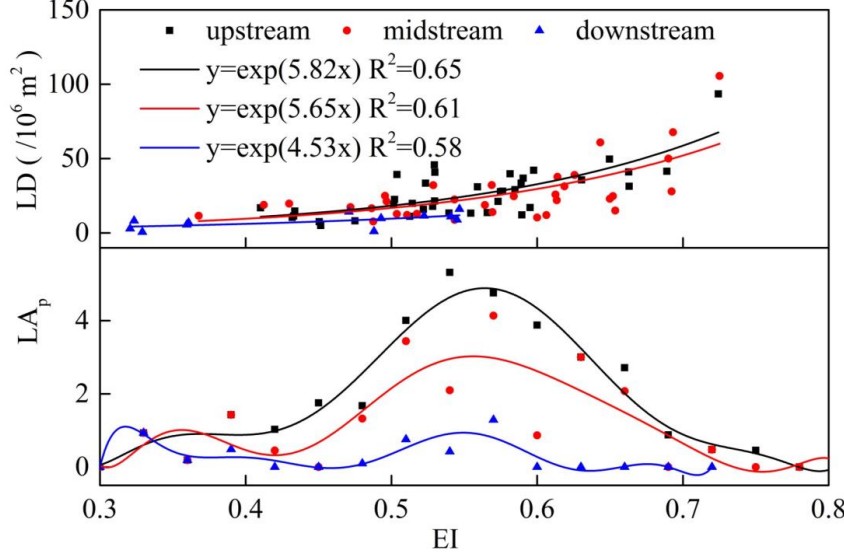


**Fig. 12** Relationship between landslides and EI in subregions.

$LA_p$ mainly increases first and then decreases as EI increases, and the $LA_p$-EI curve in the range




of less than 0.54 is higher than the range of more than 0.54, which has the similar tendency with the
$LA_p$-EI curve in all tributaries of JJG. Also the $LA_p$ in upstream and midstream is higher than upstream,
lower $LA_p$ exists in tributaries at the younger evolutionary stages. Meanwhile, lower LD and larger
$LA_p$ is in the downstream, which is at the old evolution stage, which means that with the occurrence of
historical landslides or large landslides in slope surface, the tributary has reached a stable state.
**5 Discussion**
5.1 The Power-law frequency verification of landslides distribution
Power-law frequency-magnitude relationship has been generally observed for landslides at a wide
range of size (Hovius et al., 1997; Stark and Hovius, 2001; Malamud et al., 2010), but for a small-scale
gully like JJG there is no report in literatures. For the landslides in JJG, the power law is perfectly valid
(Fig. 13), with exponent being 4.32, which differs much from the exponent for landslides over large
scale regions, such as those in the Gorkha area (2.5), the Northridge, California (2.30), and the
Wenchuan area (2.19), and many other regions (Eeckhaut et al., 2007; Lari et al., 2014). The
verification of power law confirms that the landslides area interpreted is reliable.

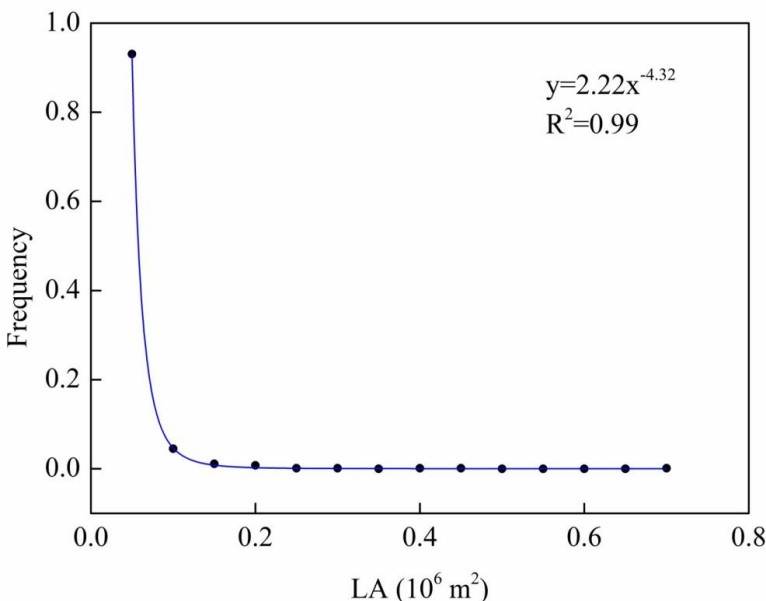


**Fig. 13** The landslide area frequency distribution of JJG.
Meanwhile, both LD and $LA_p$ are found to satisfy the Weibull distribution, with the shape
parameter and scale parameter being 1.44 and 26.86 for LD and 1.62 and 0.64 for $LA_p$, respectively
(Fig. 14). The higher scale parameter of LD suggests that LD is more sensitive to tributary distinction.





Meanwhile, the peak of frequency curve of LD is 11.86 and 0.22 of $LA_p$, which indicates that the
landslides of most tributaries are distributed on small scales.

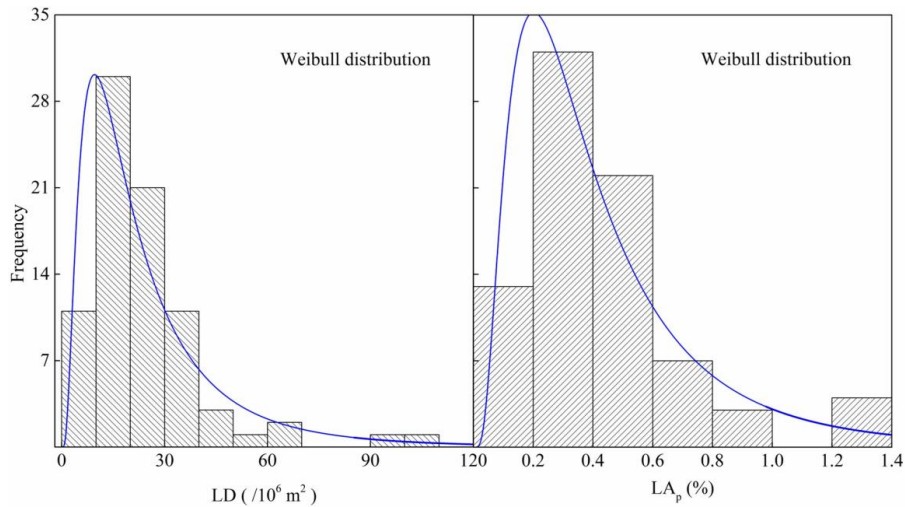


**Fig. 14** The frequency distribution of landslides among tributaries.
5.2 Distribution of EI

In JJG, EI of tributaries satisfies the Weibull distribution with scale and shape parameter of 0.58

and 6.08, this is comparable to the EI distribution of tributaries in the Wenchuan earthquake region
where the scale and shape parameter is 0.53 and 11.73, respectively (Fig. 15). The scale parameter can
reflect the EI range of variation, which varies between 0.38 and 0.64 in the Wenchuan area and
betweem 0.32 and 0.84 in JJG. The difference here can be attributed that a number of tributaries in JJG
having no landslides, while in Wenchuan, landslides distribute in almost every tributary. This also
implies that landslides occur in tributaries within a relatively narrow range of EI. More important point
is the difference between shape parameters, the bigger shape parameter in Wenchuan region means that
the curve is to the right more than in JJG, implying that the earthquake is inclined to induce more
landslides in tributaries of big EI. As JJG is of tributaries with wide range evolution stages, we choose
it as the study area to reveal the mechanism of landslides distribution.





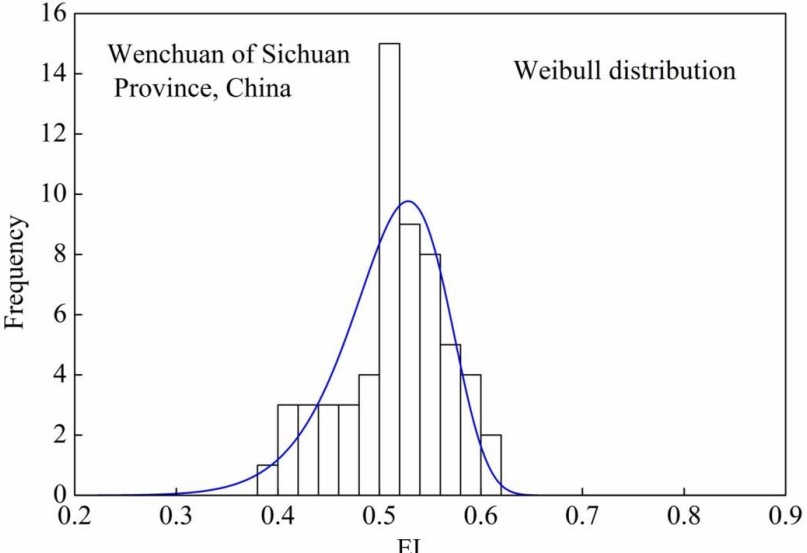


**Fig. 15** The EI frequency distribution of Wenchuan in Sichuan province.


5.3 EI and tributary morphological features


5.3.1 EI and slope distribution


As a comprehensive topography index, EI reflects the geomorphology characteristics of the
tributary. Fig. 16 shows how slope varies with EI on average, as it is crucial for landslides and debris
flow formation. The maximal average slope, usually bigger than the friction angle of the soil, occur
mainly between EI of 0.5-0.65, this coincides with range of most landslides distribution, and this also
accounts for the relationship between EI and LD which indicates that EI is related to the number or
frequency of landslides. Meanwhile, the landslides are concentrated in tributaries of class III (EI =
0.55~0.65), and these tributaries are concentrated in the midstream and upstream, mainly in the
Menqian Gully. The landslides distribution in tributaries of different EI quantitatively reveals spatial
heterogeneity distribution. The spatial distinction of landslides distribution results from the diverse
evolution stages of tributaries, which provides a heterogeneous background for material supplying in
gully. The spatial heterogeneity distribution can reveal the reason why landslides are frequent in some
tributaries while occasional in others, thus roughly to predict the landslides activity of tributaries,
which is of great significance to the comprehensive management of small watershed.





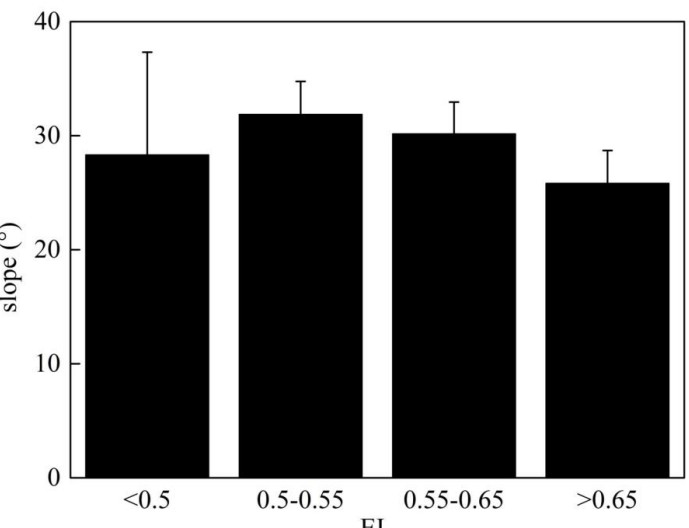


**Fig. 16** Slope variation in different evolution stages.


5.3.2 EI and channel density

Debris flow converging from tributaries into mainstream channel depends on the flow routes, or

the stream length of each tributary, and this can be described by the channel density (i.e., the length in
unit area of a region). Fig. 17 shows the density variation with EI, indicating that the channel density of
tributary is increasing as EI rises, which is conducive to the occurrence of debris flow activity.

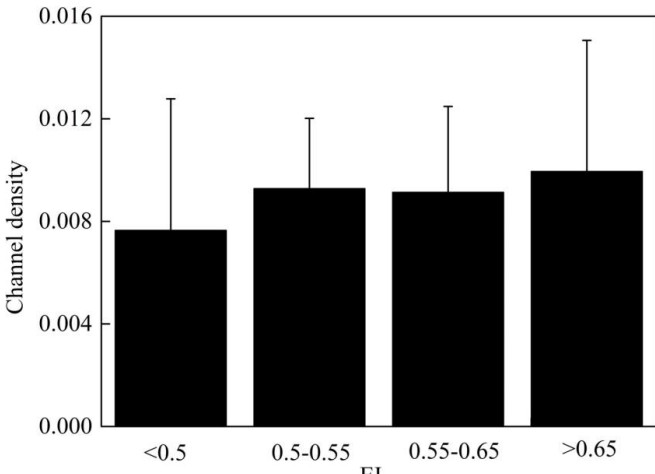


**Fig. 17** Channel density variation in different evolution stages.

Then the tributaries of EI between 0.5 and 0.65 provide favorable condition both for landslides

and flow convergence, and thus facilitate the forming and developing of debris flows.



5.4 Implication in debris flow surges


The spatial heterogeneity of tributary distribution reveals the variety of debris flow sources. As it
is difficult to observe the debris flows of each tributary, we usually see the convergence debris flows
from multiple sources. Debris flow surges always present the characteristic of diverse forms from the
perspective of material supplies (Li et al., 2015), and this can be attributed to the spatial heterogeneity
of evolution and landslides activity of tributaries as discussed above.
Previous studies usually consider debris flows activity on the gully scale and ignore the
distinction on tributary scale (Chen and Wang, 2017; Malet et al., 2004), they cannot tell the feature of
debris flows from multiple sources and undergoing diverse tributaries processes, such as initiation on
slope, flow downwards in tributary channel, and confluence into the mainstream, all closely related to
the tributary feature.
Besides, the formation of debris flow is activated by rainfall (Chen et al., 2006; Fuchu et al., 1999;
Fusco et al., 2017; Kuo and Chuan, 2007; McArdell et al., 2007; Reneau and Dietrich, 1987; Tan and
Han, 1992), different rainfall intensity and amount is in different tributaries, which adds more diversity
to the surges. The factor of precipitation will be the next study to consider and understand the
formation mechanism of debris flow surges.

**6 Conclusions**
This study has revealed the spatial heterogeneity of a debris flow gully through landslides
distribution in tributaries of different evolution stages. It is found that most landslides are distributed in
the relative young tributaries (with evolution index between 0.5 ~ 0.6). Generally, the LD increases
exponentially with EI and the $LA_p$ is concentrated in EI between 0.5 and 0.6, in accordance with the
general landslides distribution. Meanwhile, the EI satisfies the Weibull distribution, such distribution
feature also occurs in the Wenchuan area.
The majority of tributaries are at the EI range between 0.5 and 0.6, which means that sufficient
material from landslides for debris flows can be provided, which explains the reason that JJG has the
debris flow a of high frequency. In addition, the landslides distribution in JJG reveals the nonuniform
distribution of material sources for debris flows, which provides the fundamental evidence for the
variety of debris flow surges.





**Acknowledgments**

This research is supported by the National Natural Science Foundation of China (grant no.

41471011), Key International S&T Cooperation Academy of Sciences (grant no. 2016YFE0122400).

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
