# Peer review of "Landslides distribution at tributaries with different evolution stages in Jiangjia Gully, southwestern China"

_Natural Hazards and Earth System Sciences, 2019_

## Referee Comment (RC1) · Anonymous Referee #1 · 1 Aug 2019

Review of MS #nhess-2019-90 submitted by Tian and colleagues. In general, the contents of this manuscript are suitable for publication in this journal. Nevertheless, significant revisions are still needed. First of all, I suggest to clearly state the novelty of this study in abstract and introduction. In this manuscript, the authors try to demonstrate the relation between landslide density and EI (a geomorphic index). However, as mentioned, landslides may be influenced by many factors, such as geomorphology, lithology, and rainfall. Thus, I expect you to provide information about rainfall and lithology in JJG. If the other two parameters are uniform or similar, then you can directly make comparison between the landslides and the geomorphic indices. Secondly, although the authors provide a large amount of data, language needs to be greatly improved

so that the main findings of this investigations can be clearly illustrated. In particular, some sentences are too long to understand.

Text: 1. Line 32, add some most relevant reference for the EI index. 2. Line 38, Hamza et al., 2018, delete "V" and check the citation form throughout the text for consistence. 3. Line 57, you state 'Moreover, the debris flow behaviors in JJG are representative and similar phenomena are subsistent in other parts of the world', how do you know that? 4. Line 94, provide information about data source of 10 m DEM applied in this study. 5. Lines 95–96, how to divide the tributaries based on field investigation? 6. Line 135, the area of landslide is 0.38 m2? Is this really a landslide? and, how can you identify this in such a resolution? 7. Line 136, what is the meaning of accuracy here? 89.21% of the identified landslides have been observed in the field? 8. Line 147, why do you define the curves like this, and what is the meaning of this equation. 9. Line 161, scale parameter of 0.58 and shape parameter of 6.08 are inconspicuous in Figure 6. More descriptions in detail are needed. 10. Line 182, you claim that 'For a given elevation of point, larger area above it means strong slope process in the upstream'. Do you have some references or evidence to support it?

Figures: 1. I suggest to delete Figure 8, as the same spatial distribution of the landslides can be better shown in Figure 9. 2. Figures in similar pattern can be merged, such as Figures 10 and 11, Figures 14 and 15, Figures 16 and 17. 3. Figure 1, the extracted river network is not precise, especially in the downstream; what is the meaning of the dividing line of three segments, and how do you define the location of these two lines, e.g., elevation or distance from the river mouth? 4. Figure 2, where did you take the picture? Mark the location in the index map, e.g., Figure 1. 5. Add coordinates in Figures 3, 5, 8 and 9. 6. Figure 7, provide more details about definition and meaning of 'inflection points' in hypsometric curves. You mentioned 5 stages in the text. but, only 4 in the figure.

2019-90, 2019.

---

## Referee Comment (RC2) · Odin Marc (Referee) · 18 Sep 2019

Review on Tan et al.,
**Landslides distribution at tributaries with different evolution stages in Jiangjia Gully, southwestern China**

**General summary** : The author study a debris flow prone catchment. They performed some landslide mapping and some topographic classification of sub-catchments and attempt to link the two to propose that hypsometric analysis may be useful for landslide studies. The topic may be of interest for NHESS readers. Hypsometric analysis in relations to landslide are relatively rare, so it may be novel, but therefore the utility of such new index also need to be thoroughly demonstrated, that is not the case in the current state of the paper.

**Major concerns:**
1/ The quality of the writing is very variable and many sentences are hard to understand and/or highly ambiguous. Some English support and clarification would be needed before the author resubmit. The two last sentences of the conclusions (L325-329) are of these sorts.

2/ The quality of the figure caption is extremely low. A number of figures (10B, 11B, 12B and 13) contain lines (presumably fitted) without any explanation on the functional form and the fit criterium.

3/ Overall the author present various analysis, more or less standard, eg, EI ofHypsometry and inflection point in the Hypsometry, and often relate it to landslide or debris flow through bold and totally unsupported/unreferenced statement, as the few examples below:

L183: "For example, inflection points in EI between 0.45~0.55 are generally higher than those in EI below 0.45, indicating that such tributaries are more prone to landslides activities"

L186-187 : "the more concave the [hypsometric] curve is presented, and the smaller of the elevation value corresponding to the inflexion point is, which indicates that the elevation changing in unit area is small, such a tributary is not conducive to the occurrence of landslides."
(Note that these two statements are even done before presenting the result of the landslide maps...)

L223: "The major branches of JJG, the Gully of Menqian and Duozhao, are distinctive in debris flow and landslides activities."
L241: "Meanwhile, lower LD and larger $LA_P$ is in the downstream, which is at the old evolution stage, which means that with the occurrence of historical landslides or large landslides in slope surface, the tributary has reached a stable state."

All these statements make it sound as if the conclusions are known before hand (what drive landslides in which part of the study area), and the methods of the authors (EI for example) have already been largely demonstrated to be relevant to landsliding. This is not the case to my knowledge, and likely why no references are given by the authors in this type of statements. Several of these sentence are also good example of the ambiguous and unclear writing in themselves.

4/ The landslide mapping is not defined clearly enough for the readers to understand what is mapped and to what it can relate. This means again that many of the interpretations are ambiguous. I give suggestion to strengthen and clarify the mapping description in the line by line comments.

As a result my general feeling is that most of the author claim are unsupported, and that although the topic and approach may bring something of interest to the community, the paper is at the moment quite far from being at publications standard: I think that many new analysis should be performed (typically

a probability ratio or relative hazard approach (cf Milledge 2019, Rault et al 2019) with associated uncertainty level should be performed to demonstrate the utility of EI for landsliding, and a comparison with simpler classic index such as slope) and most of the paper should be rewritten/updated accordingly. This is not so far from rewriting the paper from scratch.

Below I provide a number of other criticism and suggestions:
Odin Marc

**Line by Line comments**

Introduction: It seems strange to me to put emphasis on the method of the paper rather than on its object (landslide and debris flow hazard).

L44-48 : The fact that EI merges several topograhic effect (slope, curvature , others) in an integrated index, is more likely to blurry rather than reveal underlying, physical landslide control…
However, I agree that it may contains information about disequilibrium state of subcatchment, river incision state or other long-term geomorphological effects that may have influence landsliding. But choosing a complex index the author must demonstrate that any potential link between EI and landslide is not simply due to a correlation between EI and well-know landslide

L62 63: The question of the link between EI and material supply (not clear what this is) and debris flow frequency is stated and will be repeated in conclusions but almost without any other mention in the result or discussion section (except in 5.4 under the form of a general discussion on debris flow where EI is not even mentioned). This is a problem and the paper in its current state should focus in Introduction and conclusions only on the link between EI and landslide, not debris flows.

Landslide mapping  methodology : It is overall too long but also lacking important elements :
L114: is slightly ambiguous, because no article was used. If only one image was used I would write "A or One Quickbird image was purchased to create an inventory..."

L121-122: This sentence is subjective and not bringing any information on the method. Remove.

L125: From the Fig 8 it does not seem that polygons are initiation zones of landslides but the whole affected area, including runout and deposit.

L128: This individual delineation does not seem to have been performed judging from the map in Fig 8.
To show that, only landslide outline should be shown, and preferably 2-3 landslide complex should be shown in a supplementary figure where the outline can be compared to the imagery.

L134-135 : The author should state here (or at the start of the 4.1 section) what are the landslide they mapped : recent soil or rock avalanches ? Debris flow ? Large, old deposits ? Large slope instabilities visible through deformation indicators ?
Typically small shallow disruption of the soil and vegetation may have been caused by recent storm activity, and be recovered by new vegetation in a few years, while large, undated deposits may be 100-1000 of years old, and have a very different origin. Knowing what contains the inventory is essentila

for any interpretation in terms of the landslide triggering and susceptibility, as well as for their geomorphological impact.

L136: The accuracy of what was checked by GPS ? Currently this sentence is cryptic and must be precised.

L154-160 : this refined classification is not clear in Fig 5 and does not seem to bring anything as you go back to Strahler youthful/mature distinction a few lines below (L165).

L161-163 : The preference for the Weibull distribution is not demonstrated. Anyway it is not needed for the conclusion of the author, that is that most catchments are between 0.4-0.6.

L165 : Unsupported and unexplained statement.

Section 3.3 : Very poorly written section with unclear message and no obvious support from the figure 7. Nowhere it is explain why and how EI is useful for this studies. The author just claim it indicates high or low landslide activity, without explanation or support.

L207: "It is clear that major of landslides are distributed in subregions of III and IV, with EI between 0.55 ~ 0.75."
No it is not clear. I see many landslides in II. The only way to show that EI has a substantial impact on landslide probability would be to compute the ratio between the fraction of landslide in different EI class and the fraction of the study area taken by each of this class. Thus a relative Hazard could be defined and analyzed, see the methods in Rault et al., 2019 and Milledge et al., 2019 for examples on the link between landslide and other topographic metrics. IN any case Fig 9 should be replaced by such an analysis as it will never allow to make any quantitative deduction.

Section 4.2.2 : I have the impression the author should spend more time thinking to why LD seems to increase monotonically with EI while LA does not. If this is not noise, this would suggest a very different behavior between small landslides (driving LD) and large landslides (driving LA).
The meaning of the lines in the bottom subplot of Fig 10-12 is not explained.
For me Fig 11 brings nothing : upper subplot is equal to Fig 10 and Lower subplot look just like noise.

L246 : I am quite sure Malamud 2010 does not exist. The reference in the bibliography has correct title but wrong date, should be 2004. Further, all the study you cite also clearly observed and analyzed the existence of a roll-over, that is a decay of frequency below a modal area. This does not seem to exist in your data.

L251 : "The verification of power law confirms that the landslides area interpreted is reliable."
The sentence is logically fallacious, and additionally the lack of a rollover and the very high exponent (4.32, while exponent above 3 are already rare) may suggest that the mapping is wrong or problematic. Further absolutely no information on how the exponent was obtained is given. I would suggest reading Clauset 2009 and using their method.

Section 5.2 and 5.3 : The author align a succesion of poorly explain and poorly supported statement. As a striking fact these two sections (about half the discussion) are devoid of any reference from the literature. Nothing allows the reader to evaluate /understand the discussion relative to the Wenchuan earthquake.

5.3 could have been an important section, trying to demonstrate why EI may be a better index than more classical index such as slope or flow convergence, but it is not reaching this goal yet. The method/figure are extremely naive and again I would recommend the author to go and inspire themselves from work that uses probability ratio to test whether some areas of the landscape are more or less prone to landslides ( Rault et al., 2019 and Milledge et al., 2019)

L254-258 : Landslide intensity (LD and LA) satisfy the Weibull distribution (but the author do not say how they tested and validated this claim). In any case what do the conclude from it ? I do not unerstand the link between this distribution and the claim from the author that "the landslides of most tributaries are distributed on small scales."
I am not even sure to understand what they mean.

L272 : This sentence as nothing to do here, it seems like a method comment on why you chose this study area.

The conclusion start with a summary of results, but I am unconvinced by many of them (see comments above). Then it ends with cryptic, unsupported statement. So it shouldlikely be rewritten fully after the results and method have been updated thoroughly.

Figures and Tables:
Table 1: What is the use of this table ? I do not see much use of having all the number/name of sub catchment… Clarify the utility of these numbers of remove.

Fig 2 : a lot of missing info in the caption: Where is the picture ? What are we supposed to understand from it ? Put the mean density rather than inequalities in the right plot. Does it make sense to show the sequence ? Or a mean and standard deviation ( or interquartile range) would be enough ? The mean velocity does increase but scatter and overlap remains large.

Figure 4 : Are the number in parenthesis the n and k values ? Again a Caption/Figure that cannot be understood without ambiguity.

Fig 5: What is below the catchment EI color ? Something from blue to red that is confusing and useless… remove it. I have the impression category V is not in any catchment. I do not see a xlear spatial pattern of EI within the study area

Figure 6 : Why a Weibull distribution ? What is the fit quality relative to your data ? A Normal distribution may be just as fine given your histogram.

Fig 8 : What is the background ? A satellite image ? It is very poorly visible, with half the study area covered by solid landslide polygons.

Fig 9 : Useless, both info (EI map and Landslide map) exist in other figures. Should be removed.

Also the landslide maps is problematic to me : We cannot know what the author have mapped, without images from the satellite of from the ground, and given that the mapping method is imprecise.
In any case, it seems clear to me that some polygons are likely amalgamated landslide complex rather than individual landslide, with all the problem that amalgamation can bring (see Marc and Hovius 2015).

Figure 13 : This figure is absolutely useless. As it should obviously be in Log-Log coordinate to see anything…

Fig 16 and 17 :  What is in these figures ? The mean the median ? What is the error bar meaning ? Without a statistical test, and/or some measure of uncertainty it is not clear that the difference in the different bins of EI are meaningful…

**References :**

Marc, O. and Hovius, N.: Amalgamation in landslide maps: effects and automatic detection, Nat. Hazards Earth Syst. Sci., 15(4), 723–733, doi:10.5194/nhess-15-723-2015, 2015.

Milledge, D. G., Densmore, A. L., Bellugi, D., Rosser, N. J., Watt, J., Li, G. and Oven, K. J.: Simple rules to minimise exposure to coseismic landslide hazard, Natural Hazards and Earth System Sciences, 19(4), 837–856, doi:https://doi.org/10.5194/nhess-19-837-2019, 2019.

Rault, C., Robert, A., Marc, O., Hovius, N. and Meunier, P.: Seismic and geologic controls on spatial clustering of landslides in three large earthquakes, Earth Surface Dynamics, 7(3), 829–839, doi:https://doi.org/10.5194/esurf-7-829-2019, 2019.

---

## Author Comment (AC1) · 29 Oct 2019

The lithology in this region is weak as a whole, it undergoes active neotectonic movement, faults, and folds; and rocks are dominated by slate, dolomite, limestone, basalt and breccia rocks, which are easily weathered (Gabet and Mudd, 2006). The exposed strata in this gully is mainly shallow metamorphic rocks of the lower proterozoic Kunyang group, accounting for about 80% of the gully area (Wu et al., 1990). Rainfall, as an incentive factor, mainly promotes the occurrence of landslides activity, but has little effect on the storage state of materials in the tributary. In addition, landslide susceptibility assessment is of great significance for the disaster prediction and prevention.

[Figure]

At present, most studies used statistical methods by the influence factors of landslide distribution, or based on physical models to determine the assessment result, the research of these methods is mainly focused on the gully scale. At the same time, these methods do not focus on the specific principle of material storage. In this paper, the surface erosion index, being the integral of the hypsometric curve, is adopted to explore the landslides distribution characteristics in different tributaries of the gully.

1. Line 32, add some most relevant reference for the EI index. Some relevant references have been added. 2.Line 38, Hamza et al., 2018, delete "V" and check the citation form throughout the text for consistence. The citation form throughout the text has been checked and modified. 3.Line 57, you state 'Moreover, the debris flow behaviors in JJG are representative and similar phenomena are subsistent in other parts of the world', how do you know that? Debris flow phenomena is subsistent in other parts of the world, they are intermittent, and debris flow behaviors in JJG are complete and various, so this research is carried out in JJG. 4. Line 94, provide information about data source of 10 m DEM applied in this study. This has been added in the paper, the DEM was purchased in the Sichuan surveying and mapping bureau. 5.Lines 95–96, how to divide the tributaries based on field investigation? The tributaries are divided based on field investigation result that each tributary is a complete unit for observable landslides and debris flows, also according to the fact that debris flows are prone to occur instead of direct extraction based on the same water collection threshold. In other words, these tributaries are all conspicuous in surface mass movement and loose materials are distinguishable on its slope. The tributaries are extracted from the DEM using GIS tool and also with artificial correction to ensure the accuracy of boundaries. In principle, the gully can be divided further into smaller tributaries, but that makes little difference for the present purpose as to distinguish tributaries in evolution. Some tributaries in field are displayed in Fig. 2, obviously, there are significant differences among these tributaries. 6.Line 135, the area of landslide is 0.38 m2? Is this really a landslide? and, how can you identify this in such a resolution? The data has been checked and modified. As two topological surfaces overlap and form a slight superposition, all

the topological surfaces of the landslides were checked and modified. 906 landslides have been identified, with area ranging of 2.53 × 102 ∼ 6.7 × 105 m2. 7.Line 136, what is the meaning of accuracy here? 89.21% of the identified landslides have been observed in the field? We mainly in the field observed the location of 100 landslides, and the area of some landslides we estimated. The accuracy of the landslides location and area reaches 89.21%. 8.Line 147, why do you define the curves like this, and what is the meaning of this equation. This is a fitting curve, this equation is used for fitting the curve, and the fitting effect of this equation is suitable, the fitting parameter reaches 90% above. 9.Line 161, scale parameter of 0.58 and shape parameter of 6.08 are inconspicuous in Figure 6. More descriptions in detail are needed. The small value of scale parameter means that EI is much concentrated and EI of most tributaries in JJG is mainly between 0.5 and 0.6. The shape parameter is more than 1 and the frequency of the tributaries changes rapidly with the increasing of the EI, indicating that there is a great difference among the active tributaries. According to the frequency distribution of EI, the tributaries of JJG is generally in mature and youthful evolution stages, that is the reason why high frequency debris flow occurred in JJG in the past several decades. 10.Line 182, you claim that 'For a given elevation of point, larger area above it means strong slope process in the upstream'. Do you have some references or evidence to support it? This part has been rewritten as following. For a given elevation of point, larger area above means that more material are concentrated. For example, inflection points in EI between 0.45∼0.55 are generally higher than those in EI below 0.45, indicating that more material concentrates in such tributaries, which are more prone to debris flow activities. Correspondingly, the lower the hypsometric curve is, the more concave the curve is presented, and the smaller the a/Aip is, which indicates that the elevation changing in unit area is small, such a tributary is not conducive to the occurrence of landslides and debris flow activities.

Figures: 1.I suggest to delete Figure 8, as the same spatial distribution of the landslides can be better shown in Figure 9. This is a general landslides distribution figure, and the other is landslides distribution figure in tributaries of different EI divisions. In

this paper, the general distribution of landslide is firstly written, and then the landslides distribution characteristic in different evolutionary periods is displayed, which is better to understand the structure of the paper, so these two figures are both kept. 2.Figures in similar pattern can be merged, such as Figures 10 and 11, Figures 14 and 15, Figures 16 and 17. Figures 10 and 11, Figures 16 and 17 (preceding) has been merged, now is Fig. 12 and Fig. 14, and Figures 14 represents the landslides distribution characteristic of the whole gully, and overall landslides distribution is mainly compared with other regions in the paper. Therefore, Figures 16 (preceding), representing landslides distribution characteristic of tributaries, has been deleted.

Fig. 12 Relationship between landslides and EI in subregions.

Fig. 14 The variation of the tributary morphological feature in different evolution stages. 3.Figure 1, the extracted river network is not precise, especially in the downstream; what is the meaning of the dividing line of three segments, and how do you define the location of these two lines, e.g., elevation or distance from the river mouth? The river network has been extracted and modified, as shown in Figure 1.

Fig. 1 The location of JJG. a The location of Dongchuan in China. b Dongchuan Debris Flow Observation and research stations. c Deposition of surges.

The line of the three segments divided is based on the difference of elevation, the three segments is upstream, midstream and downstream of the gully, respectively. 4. Figure 2, where did you take the picture? Mark the location in the index map, e.g., Figure 1. This picture was taken in the field survey, it is in the Menqian gully, close to the mouth of the Menqian gully and Duozhao gully, the location is shown in the following figure and indicated by the yellow arrow. The phenomenon can be found in other location of the gully, this is just one of them and is used to illustrate the phenomenon of debris flow surges and variety of material accumulation.

Fig The location of debris flow surges and variety of material accumulation 5. Add coordinates in Figures 3, 5, 8 and 9. 6. Figure 7, provide more details about definition and

meaning of 'inflection points' in hypsometric curves. You mentioned 5 stages in the text. but, only 4 in the figure. All figures have been remade, and the coordinates have been added in the figures. More details about definition and meaning of 'inflection points'in hypsometric curves have been provided in the following. Obviously, the hypsometric curves exhibit different shapes, which can be featured by the inflection point, defined as the zero point of the second derivative of the fitting curve (Eq. 2): ïijĹ2ïijĽ where x denotes a/A, and y=0 determines the inflection point at a/Aip. It is found that the a/Aip varies with EI in a power law form (Fig. 6), meaning that the bigger the evolution index is, the lower the inflection points of the curves are. The higher the EI value the lower the inflection point, and this implies that there should be more material accumulated in the lower part of the tributary, which are relatively easy to join the the debris flow.

Fig. 6 The relationship between the inflection point and and EI. Moreover, we display the hypsometric curves of different evolution stages in Fig. 7; in particular, the inflection points of the curves (the rectangle in each plot) are displayed in different position of the curves. The inflection point indicates the elevation of a tributary with area varing. It can be seen in Fig. 7, which shows that the larger of the EI is, the smaller of the a/Aip is. When the point is high, the changing occurs at the high elevation, i.e., mainly in the upstream of the tributary. Since there is no evolution area more than 0.75 in JJG, four major evolution divisions are analyzed.

Fig. 7 Hypsometric curves of different EI divisions The evolution curve changes from concave to convex with the increasing of evolution value, and the convex form of the tributary is more conductive to the material movement of the tributary and more loose materials are produced. For a given elevation of point, larger area above means that more material are concentrated. For example, inflection points in EI between 0.45∼0.55 are generally higher than those in EI below 0.45, indicating that more material concentrates in such tributaries, which are more prone to debris flow activities. Correspondingly, the lower the hypsometric curve is, the more concave the curve is presented, and the smaller the a/Aip is, which indicates that the elevation changing in

unit area is small, such a tributary is not conductive to the occurrence of landslides and debris flow activities. Some landslides distribution of tributary in different evolutionary periods is shown in Fig. 8. In the tributary within the EI range of 0.35-0.45, the landslides distribution is scattered with the large area and low number, and the tributary is generally concave, which is not conductive to the materials movement. In addition, with the increasing of the evolutionary value, the landslides number is increasing and the area is decreasing, and the tributary in high EI division is convex, which is conductive to the materials movement.

Fig. 8 Some landslides distribution tributaries of different EI divisions

Please also note the supplement to this comment:
https://www.nat-hazards-earth-syst-sci-discuss.net/nhess-2019-90/nhess-2019-90-AC1-supplement.pdf
* * *
The plot area (upper-left panel):

LD ($/10^6$ m$^2$) vs EI

- ■ Menqian Gully
- ● Duozhao Gully
- —— $y=0.37*1195.57^x$, $R^2=0.98$
- —— $y=0.07*17493.68^x$, $R^2=0.89$

(upper-right panel):

- ■ Upstream ● Midstream ▲ Downstream
- —— $y=\exp(13.19-40.79x+39.79x^2)$, $R^2=0.69$
- —— $y=\exp(15.86-51.01x+49.34x^2)$, $R^2=0.71$
- —— $y=\exp(-19.38+80.69x-72.37x^2)$, $R^2=0.74$

(lower-left panel):

LA$_p$ (%) vs EI

- ■ Menqian Gully
- ● Duozhao Gully

(lower-right panel):

- ■ Upstream ● Midstream ▲ Downstream

**Fig. 1.**

[Figure]

**Fig. 2.**

Downstream

Midstream

Upstream

Menqian Gully

Duozhao Gully

**Legend**
— Rivers
Elevation (m)
High : 3192.73

Low : 1040.52

☐ Three elevation segments

🚩 Dongchuan Debris Flow Observation and Reserch Station

0  .75  1.5      3
━━━━━━━━━━━ km

a

Beijing

Dongchuan

Legend
— Rivers
▨ Provinces

b

c

**Fig. 3.**

**Fig. 4.**

$y=0.04x^{-5.24}$

$R^2=0.74$

**Fig. 5.**

Fig. 6.

**Legend**

Landslides

EI division

<0.35

0.35-0.45

0.45-0.55

0.55-0.65

>0.65

Elevation (m)

High : 3192.73

Low : 1040.52

0    .200000000.400000000
Miles

**Fig. 7.**
